# Molecular and Cellular Bases of Immunosenescence, Inflammation, and Cardiovascular Complications Mimicking “Inflammaging” in Patients with Systemic Lupus Erythematosus

**DOI:** 10.3390/ijms20163878

**Published:** 2019-08-09

**Authors:** Chang-Youh Tsai, Chieh-Yu Shen, Hsien-Tzung Liao, Ko-Jen Li, Hui-Ting Lee, Cheng-Shiun Lu, Cheng-Han Wu, Yu-Min Kuo, Song-Chou Hsieh, Chia-Li Yu

**Affiliations:** 1Division of Allergy, Immunology & Rheumatology, Taipei Veterans General Hospital & National Yang-Ming University, #201 Sec 2, Shih-Pai Road, Taipei 11217, Taiwan; 2Institute of Clinical Medicine, National Taiwan University College of Medicine, #7 Chung-Shan South Road, Taipei 10002, Taiwan; 3Department of Internal Medicine, National Taiwan University Hospital, #7 Chung-Shan South Road, Taipei 10002, Taiwan; 4Section of Allergy, Immunology & Rheumatology, MacKay Memorial Hospital, #92 Section 2, Chung-Shan North Road, Taipei 10449, Taiwan

**Keywords:** systemic lupus erythematosus, immunosenescence, inflammaging, oxidative stress, nitrosative stress, bioenergetics, immunometabolism, advanced glycation end product

## Abstract

Systemic lupus erythematosus (SLE) is an archetype of systemic autoimmune disease, characterized by the presence of diverse autoantibodies and chronic inflammation. There are multiple factors involved in lupus pathogenesis, including genetic/epigenetic predisposition, sexual hormone imbalance, environmental stimulants, mental/psychological stresses, and undefined events. Recently, many authors noted that “inflammaging”, consisting of immunosenescence and inflammation, is a common feature in aging people and patients with SLE. It is conceivable that chronic oxidative stresses originating from mitochondrial dysfunction, defective bioenergetics, abnormal immunometabolism, and premature telomere erosion may accelerate immune cell senescence in patients with SLE. The mitochondrial dysfunctions in SLE have been extensively investigated in recent years. The molecular basis of normoglycemic metabolic syndrome has been found to be relevant to the production of advanced glycosylated and nitrosative end products. Besides, immunosenescence, autoimmunity, endothelial cell damage, and decreased tissue regeneration could be the results of premature telomere erosion in patients with SLE. Herein, the molecular and cellular bases of inflammaging and cardiovascular complications in SLE patients will be extensively reviewed from the aspects of mitochondrial dysfunctions, abnormal bioenergetics/immunometabolism, and telomere/telomerase disequilibrium.

## 1. Introduction

Diverse dysregulation in innate [1,2,3] and adaptive [4,5] immune systems has been found in patients with systemic lupus erythematosus (SLE). These kinds of immune dysregulations stimulate protean autoantibody productions [6,7] and chronic immune-mediated inflammation [1,2,3,4]. Aging is defined as the physiologically progressive degeneration and decay of somatic tissues/organs and immune systems (immunosenescence). The tissue/organ degeneration results in neoantigen formation and immune reactivity to these newly formed, “altered”, self-antigens after loss of self-tolerance [8]. Eventually, elderly people experience low-grade systemic inflammation [9], low titer autoantibody productions, and immunosenescence, mimicking autoimmune diseases [8,9,10,11]. Accordingly, the presence of immunosenescence and inflammation simulating “inflammaging” become the characteristic immune dysfunctions in both SLE patients and the elderly [8,12,13]. The similarity in immune dysfunctions and common clinical manifestations in age-associated physiological senescence and SLE are listed in Table 1 [14,15,16,17,18,19,20,21,22,23,24,25,26,27].

Complex etiological factors are involved in lupus pathogenesis, such as genetic predisposition [28,29,30,31], epigenetic post-transcriptional regulation [32,33,34,35,36], sex hormone imbalance [37,38], environmental stimulation [39,40], mental/psychological stresses [38], and other undefined factors [41]. Recently, many authors have demonstrated that mitochondrial dysfunctions [42,43,44,45,46], defective bioenergetics in immune cells, and abnormal immunometabolism [47,48] can induce structural and functional changes of biomolecules (proteins, lipids, nucleic acids, and glycoproteins) by oxidative and nitrosative stresses [49,50]. All of these precipitating factors can potentially accelerate immunosenescence by way of inflammaging in patients with SLE [8,51,52]. In addition, disequilibrium in the telomere/telomerase system has also been found to be closely related to cell senescence [53,54,55] and decreased bone marrow-derived mesenchymal stem cell (BM-MSC) production in SLE [56]. The interactions among these upstream etiopathogenetic factors in diverse immune dysfunctions in patients with SLE are rather intriguing. In this review, we are going to discuss in detail the molecular and cellular bases of inflammaging, cardiovascular complications related to mitochondrial dysfunctions, abnormal bioenergetics/immunometabolism, and disequilibrium in the telomere/telomerase system in patients with SLE.

## 2. The Cellular Basis of Inflammaging in Patients with SLE

Immunosenescence is defined as a decreased ability of the immune system to respond to foreign antigens and the inability to maintain self-tolerance. The situation may lead to an increased susceptibility to infections, cancer, autoimmunity, and chronic inflammation [14,57]. “Inflammaging” is characterized by persistent low-grade chronic inflammation in the aging process that increases plasma levels of inflammatory cytokines, low titer of autoantibodies, acute phase proteins, and coagulation factors [58]. Accordingly, it is considered that immunosenescence and inflammaging are friends as two sides of the same coin [13]. Larbi et al. [59] have demonstrated that decreases in the expression and functions of the T-cell receptor (TCR) and its co-receptors to process antigens are closely associated with immunosenescence, presenting as increased susceptibilities to infections. Bulati et al. [60] have found that inflammaging exerts a strong impact on B cell lymphopoiesis in the bone marrow and mature B cell remodeling by shifting them from naïve to memory B. Consequently, the produced immunoglobulins are defective in affinity to protect against newly encountered antigens. Furthermore, memory B cell differentiation to plasma cell is also impaired by inflammaging. Therefore, high affinity antibodies to protect the body are lacking. Analysis of immunosenescent biomarkers in patients with SLE revealed that increased CD4^+^CD28^null^ angiogenic T cell subset containing granzyme B, perforin, and interferon (IFN)-γ was closely related to cardiovascular complications, just like in the aging group [61,62,63]. In addition, increase in the IL-6/TGF-β ratio drives T cell response toward IL-22 overexpression and IL-17 maturation in patients with SLE [64,65]. Obviously, aging and SLE share the same immunosenescence, inflammation, and beyond [66]. The cellular bases of inflammaging and its pathophysiological effects on patients with SLE are summarized in Table 2.

## 3. The Factors Contributing to Inflammaging and Cardiovascular Morbidities in Patients with SLE

Although the real mechanism for inflammaging remains unclear, some precipitating factors—including oxidative-inflammatory stresses, cytokines, DNA damage, autophagy/mitophagy dysfunctions, and stem cell aging—have been postulated [67], among which, oxidative and inflammatory stresses are crucial for immunosenescence in the mouse model of premature aging [68] and patients with SLE [8,69,70,71]. The oxidative stresses are derived from free radicals containing reactive oxygen species (ROS), reactive nitrogen species (RNS), and their glycation/nitrosation end products of biomolecules (proteins, lipids, glycoproteins, and nucleic acids) [57]. Three major biochemical mechanisms linking oxidative stresses to inflammaging have been found: (1) Oxidative damage of the intracellular biomolecules impairs cell functions, (2) excessive production of oxidized molecules elicits cell apoptosis, and (3) modification of self-antigens induces neoantigen formation. Obviously, free radical production of the body is derived mainly from mitochondrial dysfunctions and defective bioenergetics/immunometabolism in the immune-related cells. The major factors that contribute to inflammaging and metabolic syndrome in patients with SLE are depicted in Figure 1.

## 4. Mitochondrial Dysfunction is One of the Crucial Up-Stream Factors in Inducing Oxidative Stresses and Immunosenescence in Patients with SLE

Mitochondria are ancient organelles essential for heat production, calcium storage, reduction-oxidation (redox) regulation, biosynthesis, cell signaling, and cell apoptosis, as well as physiologic processes against aging. Many of the functions decline with age [51]. SLE is characterized by abnormal T-cell activation, followed by its suicide death, presumably because of the over production of reactive oxygen species (ROS) in mitochondria. SLE-T cell exhibits persistently high mitochondrial transmembrane potential (mitochondrial hyperpolarization) and depletion of ATP and glutathione (GSH) that lead to activation-induced cell death [72]. Consequent to the defective apoptotic cell clearance capacity, which is associated with secondary cell necrosis, the released cellular components stimulate Toll-like receptors to result in inflammation in patients with SLE [1]. Besides, mitochondrial (mt)DNA damage [45,73,74,75,76], characterized by D-310 heteroplasmy and copy number decrease [42,77], is also present in SLE. Su et al. [78] investigated the caspase-dependent mitochondrial apoptotic pathway in immune cells from SLE patients. They found that peripheral blood mononuclear cells (PBMCs) and polymorphonuclear neutrophils (PMNs) from SLE patients exhibited increased expression of apoptosis initiators, caspase 9 and 10, that are positively correlated to the expression of mitochondrial anti-viral signaling protein and interferon regulatory factor 7. Lee et al. [42,77] further investigated sequence variations of mtDNA in SLE leukocytes. They found that decreased mtDNA copy number and increased D310 heteroplasmy are correlated to lupus nephritis. In addition, the same group have also revealed that the D-310-4977 deletion in oxidative DNA and mtDNA are also correlated to the high serum levels of cytokines (IL-10, IFN-α, IL-23, IFN-γ) and chemokines (IP-10 and MCP-1) [45].

It is believed that ROS can potentially damage intra- and extra-cellular biomolecules, including DNA, RNA, lipids, and proteins. The formation of 8-hydroxy-2′-deoxyguanosine (8-OHdG) is the most common biomarker for oxidative DNA damage. Lee et al. [79] demonstrated first an increased plasma 8-OHdG and a decreased mRNA expression of human 8-oxogunine DNA glycosylase 1 (*hOGG*1), anti-oxidant enzymes, mitochondrial biogenesis-related proteins, and glycolytic enzymes in SLE-leukocytes. The same group also showed the role of *hOGG*1 C1245G polymorphism in contribution to the susceptibility to lupus nephritis [42,80]. On the other hand, Yang et al. [81] found that urinary neutrophil gelatinase-associated lipocalin (NGAL), a 25kDa small protein belonging to the lipocalin protein superfamily, could become a potential biomarker for renal damage in patients with SLE. Recently, Lee et al. [82] further demonstrated that SLE-PBMC displayed low basal mitochondrial oxygen consumption rate and extracellular acidification rate that could be recovered by glutamine supplement. Putting these results together, mitochondrial dysfunctions in immune cells of SLE patients—including decreased redox capacity, increased oxidative-inflammatory stresses, cell apoptosis, mtDNA damage and breakdown of itself (mitophagy), and defective bioenergetics—may facilitate immune cell senescence in patients with SLE as shown in Figure 2.

## 5. Defective Bioenergetics/Immunometabolism Allied with Mitochondrial Dysfunction Accelerates Metabolic Syndrome and Immunosenescence in Patients with SLE

Reprogrammable cellular metabolism occurs in immune cells for controlling cell proliferation and differentiation. Resting lymphocytes generate energy via oxidative phosphorylation and fatty acid oxidation, whereas activated lymphocytes rapidly shift to glycolysis that requires aerobic condition [83]. The aerobic glycolysis may confer a higher rate of ATP production to maintain redox homeostasis [84]. In autoimmunity, failure in metabolic reprogramming at critical checkpoints may lead to immune cell hyper-reactivity, increase in oxidative stress, and subsequent metabolic abnormalities [48,85,86,87].

Li et al. [47] found that a defective expression of facilitative glucose transporter (GLUT)-3 and GLUT-6 in T cells and PMNs increased intracellular basal lactate but decreased ATP production. In addition, the same authors noted that the critical molecules for maintaining intracellular redox capacity such as intracellular GSH, GSH-peroxidase, and γ-glutamyl- transpeptidase were decreased in SLE-T cells. However, Choi et al. [88] found that mammalian target of rapamycin (mTOR) activation in SLE-T cells could enhance GLUT activity and TCR signaling. Rhoads et al. [89] further demonstrated that increased GLUT-1 expression in CD4 T cells, insulin resistance, and fasting insulin levels were the major factors for metabolic abnormalities in patients with SLE. The different immunometabolic abnormalities can induce immunosenescence and metabolic syndrome in patients with SLE. Ugarte-Gil et al. [90] found that SLE patients exhibited a lower percentage of naïve CD4+T cells and a higher percentage of memory CD4^+^T cells, and therefore an increase in memory CD4^+^T cell/naïve CD4^+^T ratio, which was thus closely related with their metabolic syndrome and cardiovascular morbidity. Perl [91] demonstrated that the mTOR complex (mTORC) in T cells integrated environmental cues, energy, and nutrient levels in response to metabolic status for maintaining homeostasis. It is quite interesting that Th1 and Th2 cells require mTORC1 activation, whereas mTORC2 activation only favors Th2 differentiation. Femandez et al. [92] reported mTORC1 activation in CD4^+^T cells of SLE patients. The same group also demonstrated that mTORC1 and mTORC2 could promote the transcription of genes involved in carbohydrate metabolism and lipogenesis, enhance protein translation, and inhibit autophagy [93]. Accordingly, mTOR activation can become a biomarker and a critical pathway to autoimmune disorders, cancer, obesity, and aging [93].

The cause of decreased GSH levels in SLE immune cells may result from mitochondrial hyperpolarization and increased reactive free radicals, as reported by Huang and Perl [94]. Lai et al. [95] reported that N-acetylcystein, a GSH precursor, could reduce lupus disease activity by blocking mTOR in T cells. Perl et al. [96] further demonstrated in comprehensive metabolomics analyses that the accumulation of kynurenine in response to N-acetylcysteine in patients with SLE could become a biomarker for mTOR activation. In addition, rapamycin, the original mTOR blocker from which this adaptor protein was named, has been proven effective in the treatment of intractable SLE resistant to, or intolerant of, conventional immunosuppressants as reported by Lai et al. [97]. Furthermore, Duval et al. [98] demonstrated that rapamycin treatment ameliorated age-related accumulation of toxic metabolites in brains of the mouse model of Down syndrome and aging.

In short, mTOR activation can determine the metabolic pathway in CD4^+^T cells of SLE patients to induce metabolic syndrome. These results are summarized in Figure 3.

## 6. Metabolomic Signatures Predisposing to the Metabolic Syndrome in Patients with SLE

Metabolomic study is the identification and quantification of all low-molecular weight metabolites (MW < 1–1.5 kDa) in a biological specimen. Li et al. [99] sought to identify candidates for metabolic biomarkers by gas chromatography-mass spectrophotometry (GC-MS). Twenty-five metabolites—including glutamic acid, urea, tyrosine, phosphate, and glycerol—are considered as the metabolomic signatures for SLE. Among these, elevated serum levels of glutamic acid have been found to be significantly correlated to lupus pathophysiology, as shown in the left panel of Figure 3. Furthermore, by using nuclear magnetic resonance (NMR) technology, Ouyang et al. [100] have showed a reduced level of most amino acids, glycolytic and tricarboxylic acid (TCA) cycle metabolites, and fatty acids, as well as products of oxidative stress, among approximately 30,000 metabolites in patients with SLE. In addition, a distinct pattern of serum lipids with pro-inflammatory properties, including high levels of low-density lipoprotein (LDL) and very low density lipoprotein (VLDL), as well as low levels of high-density lipoprotein (HDL), was found. These abnormal lipid profiles imply that high risks of cardiovascular complications are resulted from oxidized LDL particles in SLE [101,102,103]. Figure 3 (left panel) shows the metabolomic signatures, predisposing to the metabolic syndrome with cardiovascular complications, in patients with SLE.

## 7. Activation of AGE-RAGE System Induces Skin Autofluorescence, Cardiovascular Morbidity, and Pro-Inflammatory IL-6 Production in Patients with SLE

Advanced glycation end products (AGEs) are a class of glycosylated biomolecules (proteins, lipids, nucleic acids, or glycoproteins) synthesized by non-enzymatic incorporation of saccharides into biomolecules. The glycosylation of biomolecules is mediated by Maillard reaction, including slow formation of intermediary Schiff bases, reversible Amadori products, and, finally, the generation of irreversible molecules harmful to the body [104]. de Leeuw et al. [105] found increased AGEs in SLE were associated with the development of accelerated atherosclerosis. Nienhuis et al. [106] demonstrated that skin autofluorescence originated from cutaneous accumulation of AGE in SLE was not necessarily reflected by elevated plasma AGE levels. In contrast, Meerwaldt et al. [107] have concluded that skin autofluorescence is a strong independent predictor of, and contributor to, mortality in SLE with end-stage renal disease (ESRD).

The receptor for AGE (RAGE) belongs to the immunoglobulin superfamily of trans-membrane cell surface molecules. The RAGE also binds to other ligands, including high mobility group-box protein 1 (HMGB1), S100/calgranulins, and amyloid fibrils [108]. Binding of these ligands to the RAGE can provoke inflammatory responses. Pan et al. [109] have found that HMGB1 is associated with disease activity in SLE. In contrast, plasma levels of soluble RAGE are negatively correlated to SLE disease activity [110,111]. Martens et al. [112] have shown that RAGE polymorphisms are associated with susceptibility to SLE and lupus nephritis.

Our recent report revealed that AGE derived from bovine serum albumin enhanced monocyte IL-6 gene expression via MAPK-ERK and MyD88-transduced NF-κB *p*50 signaling pathways. These results may indicate that protein glycation end products can induce pro-inflammatory cytokine production and endothelial cell damage [113].

The involvement of AGE-RAGE activation in skin autofluorescence, metabolic syndrome, and pro-inflammatory cytokine production in SLE is shown in Figure 4.

## 8. Molecular Basis of Oxidative and Nitrosative Stresses in Inducing Autoimmunity and Cardiovascular Morbidity in Patients with SLE

Oxidative post-translational modifications of serum and intracellular biomolecules not only alter the self-antigens, but induce autoimmunity [114]. The modifications of biomolecules by oxidative stress result in the glycation and nitrosation of proteins [49], lipid peroxidation [50], and mtDNA strand break [115]. The combination of oxidant stress, inflammation, and subsequent production of glycosylative and nitrosative end products increases autoimmune reactions and cardiovascular morbidities in patients with SLE [51]. We shall discuss the roles of ROS and RNS separately in lupus pathophysiology.

### 8.1. ROS-Induced Lipid Peroxidation and Histone Modifications Play a Role in Autoimmunity and Cardiovascular Morbidities in SLE

Increased free radical production, including ROS and RNS, is mainly derived from mitochondrial dysfunctions and inflammation. Kurien et al. [116] demonstrated that increased oxidative stress-modified DNA (8-OHdG and 8-oxoguanine) and LDL could elicit premature atherosclerosis in patients with SLE. In addition, immunization with 4-hydoxy-2-nonenal-modified 60-kDa Ro autoantigen accelerated epitope spreading in an animal model of SLE. Besides, protein glycation and oxidative stress can attack poly-unsaturated fatty acid by a process of lipid peroxidation, which ultimately results in malondialdehyde, 4-hydroxy-2-nonenal modified autoantigen, and other toxic by-products. Furthermore, Kurien BT and Scofield RH [50] have demonstrated that conjugated dienes, malondialdehyde, and 8-isoprostaglandin F2α are significantly elevated in the serum of SLE patients. In addition, Mir et al. [117] reported that glycosylation of N-terminal region of a variable length in histone proteins can induce various DNA syntheses or repressions, including gene activation, gene silencing, DNA replication, and even gene damage in different autoimmune diseases. These results suggest that ROS not only mediate lipid peroxidation to elicit cardiovascular morbidities (left panel of Figure 5), but also modify self-antigens to neoantigens to induce autoantibody productions (right panel of Figure 5) in patients with SLE.

In short, oxidative stress represents an imbalance between the production of ROS and the ability to clear these reactive intermediates. In humans, increased oxidative stress is involved in many diseases, including atherosclerosis, myocardial infarction, and autoimmune responses, in patients with SLE [51,52,118].

### 8.2. Role of Nitrosative Stress in Neoantigen Formation and Autoantibody Production in Patients with SLE

Nitrosative stress is induced by reactive nitrogen species (RNS) that include nitric oxide (NO^−^), peroxynitrite (ONOO^−^), and nitrogen dioxide radical (NO_2_^●−^). Peroxynitrite is a reactive oxidant created from both NO^−^ and superoxide anion (O_2_^●−^). Accordingly, peroxynitrite is a mixture of oxidant and nitrative molecules in response to oxidative stress. The oxidant species can modify a great number of biomolecules. Nitrative stress is presumed to induce amino acid nitration, leading to formation of 3-nitrosative tyrosine [119,120,121,122]. Ahmad et al. [123] and Khan et al. [124] reported that peroxynitrite-modified histone exhibited strong immunogenicity to induce high titer anti-histone antibodies. The anti-dsDNA antibodies produced in SLE may co-localize the same antigen with anti-histone antibodies by cross-reactivity. These results indicate that peroxynitrite-modified biomolecules play a role in autoantibody production in SLE. In addition, Arif et al. [125] and Ahmad et al. [126] confirmed that anti-dsDNA-positive IgG obtained from SLE sera show a binding affinity with peroxynitrite-modified human serum albumin. These data also suggest compellingly that antibodies against peroxynitrite-modified biomolecules are implicated in lupus pathogenesis (right panel of Figure 5). In addition, DNA strand break induced by free radicals may accelerate cell apoptosis and even cell necrosis in patients with SLE.

Putting 8.1 and 8.2 together, the intricate molecular basis of increased oxidative and nitrosative stresses in lupus pathogenesis is depicted in Figure 5.

## 9. Molecular and Cellular Mechanisms Underlying Telomere/Telomerase Disequilibrium in Inducing Autoimmunity, Immunosenescence, and Bone Marrow-Derived Mesenchymal Stem Cell (BM-MSC) Senescence in Patients with SLE

Telomeres are specialized loop-like terminal chromosomal structures composed of DNA and DNA-binding proteins that cap and protect the end of chromosomes [127,128]. Telomeres progressively shorten for about 50–100 bp after each round of cell division, unless compensated by telomerase [129]. Accordingly, telomeres act like a “cell clock” and are implicated in cellular senescence in aging and SLE [48]. Telomerase, as a telomere regulatory enzyme, is inducible in T cell stimulated with anti-CD3 or phorbol myristate acetate [130]. Telomeric erosion can induce cell senescence and later apoptosis. Honda et al. [131] have demonstrated for the first time that SLE lymphocyte has shorter telomeres and a reduced replicative potential compared to the controls. Later, many authors showed similar results in peripheral T and B cells in patients with SLE [53,54,55,132,133,134,135]. Wu et al. [136] further demonstrated that premature telomere shortening in PMNs from patients with SLE was correlated with lupus disease activity. Georgin-Lavialle et al. [135] have reported that oxidative stress, inflammation, and increased leukocyte renewal are the major precipitating factors in accelerating telomere shortening. Zhou et al. [137] noted that the expression of shelterin complex *TPP1*, *TIN2*, *POT1,* and telomere maintenance protein *KU80* were significantly reduced, whereas shelterin complex TRF2 and maintenance protein MREII increased conversely in SLE patients. They deduced that the abnormal gene expression of shelterin complex and the telomere maintenance proteins might play a role in lupus pathogenesis. de Punder et al. [138] elucidated the mechanistic pathways linking telomerase activity and cellular aging in the immune system. Tan et al. [139] further confirmed that increased PTEN/Akt-P27 (kip 1) signaling promoted BM-MSC senescence and apoptosis in SLE patients. These alterations also decrease Treg/Th17 ratio to induce autoimmunity. Figure 6 illustrates the molecular and cellular mechanisms of telomere/telomerase disequilibrium in inducing immunosenescence, decreasing Treg/Th17 ratio, impairing tissue regeneration, and increasing cardiovascular complications in patients with SLE.

## 10. Conclusions

In addition to genetic/epigenetic, environmental, hormonal, and physical/psychological impactions, mitochondrial dysfunctions, defective bioenergetics/immunometabolism, and telomere/telomerase disequilibrium also become crucial factors for inflammaging in patients with SLE. The key molecules for inflammaging are reactive oxidant molecules, including ROS and RNS, that elicit powerful oxidative and nitrosative stresses. As a consequence, immune dysregulation and chronic low-grade inflammation ensues. In addition, defective bioenergetics/immunometabolism further facilitate the production of AGEs, nitrogen radicals (peroxynitrite), and lipid peroxidation that cause metabolic syndrome and cardiovascular co-morbidities. In addition to telomere/telomerase disequilibrium causing premature immune cell senescence, BM-MSC also impairs tissue regeneration in patients with SLE. Anti-oxidant therapy may likely become a supplementary therapeutic modality for active SLE in the future.

## Figures and Tables

**Figure 1 ijms-20-03878-f001:**
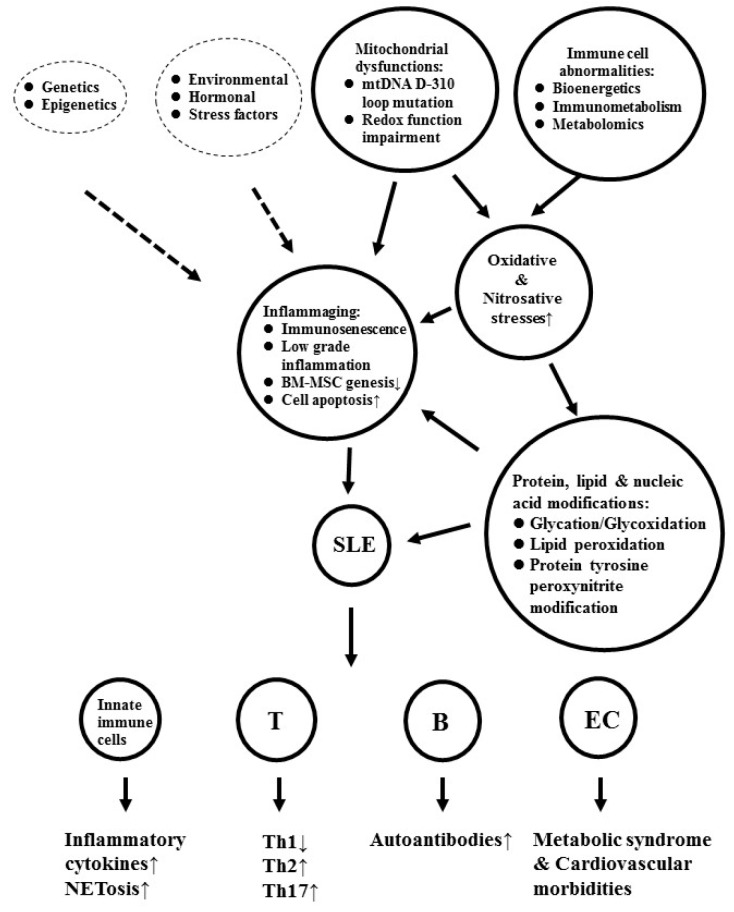
The factors that contribute to inflammaging and metabolic syndrome in patients with SLE. The broken circles are not discussed in the present review. mt: Mitochondrial; Redox: Reduction and oxidation; BM-MSC: Bone marrow derived mesenchymal stem cell; T: T lymphocyte; B: B lymphocyte; EC: Endothelial cell; Th: Helper T cell; NETosis: Apoptosis of neutrophil to form neutrophil extracellular trap.

**Figure 2 ijms-20-03878-f002:**
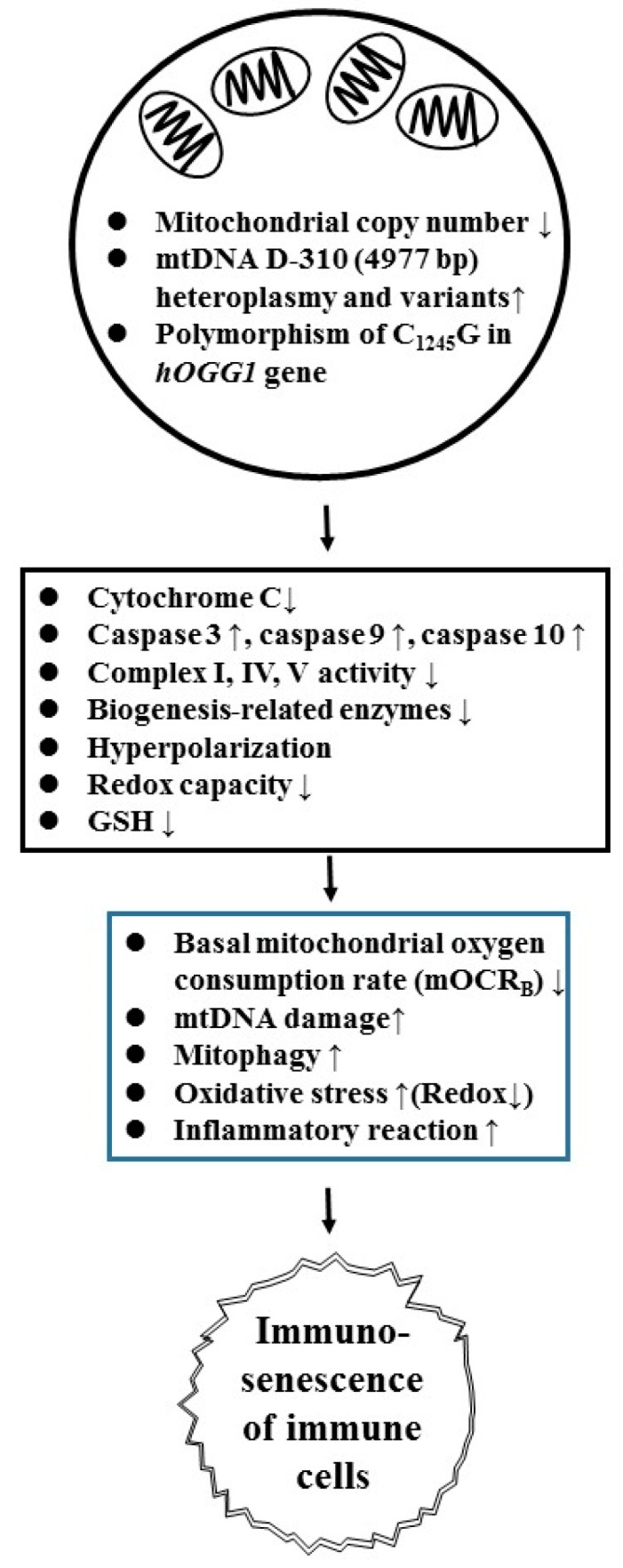
The molecular bases of mitochondrial dysfunction in inducing immune cell senescence in patients with SLE. mt: Mitochondrial; *hOGG1*: Gene encoding human 8-oxoguanine DNA N-glycosylase 1; Redox: Reduction and oxidation.

**Figure 3 ijms-20-03878-f003:**
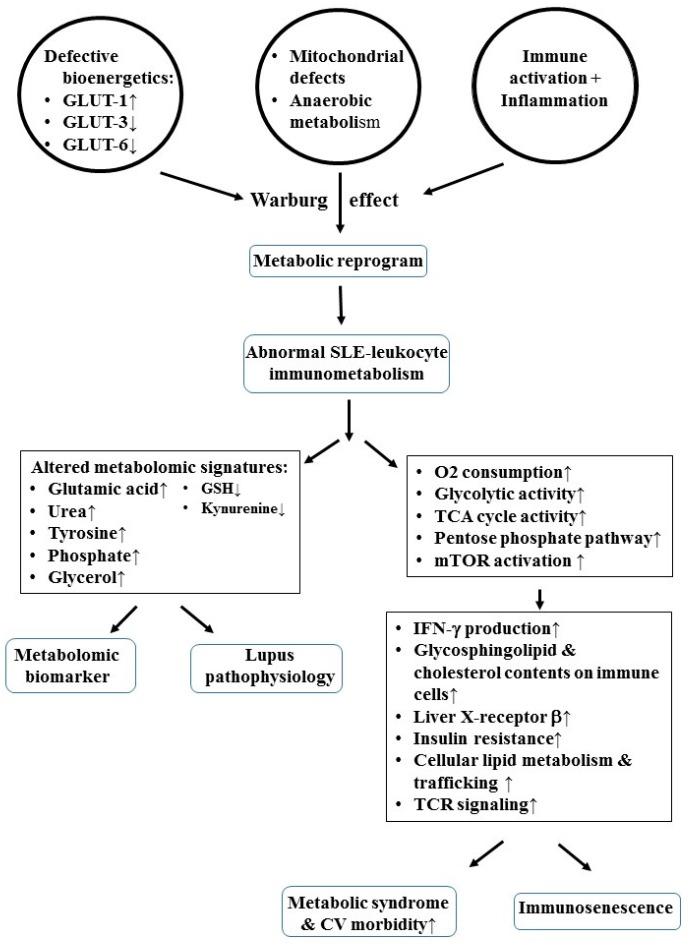
Defective bioenergetics/immunometabolism in association with mitochondrial dysfunctions mark the metabolomic signatures and predispose metabolic syndrome in patients with SLE. GLUT: Glucose transporter; TCA: Tricarboxylic acid; mTOR: Mammalian target of rapamycin; IFN: Interferon; TCR: T cell receptor; CV: cardiovascular.

**Figure 4 ijms-20-03878-f004:**
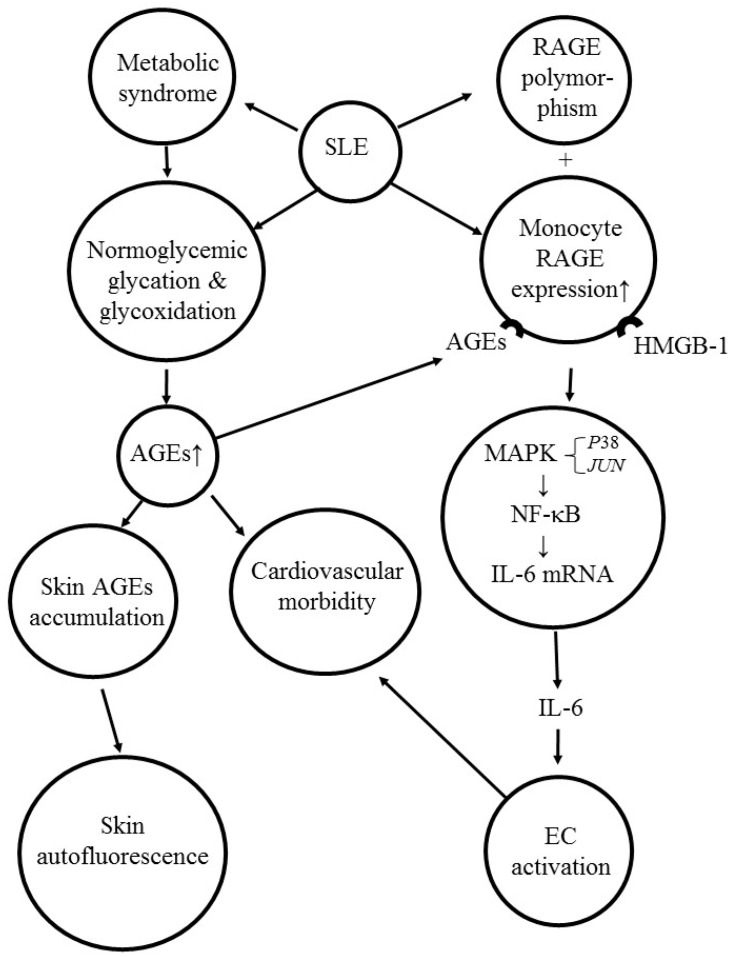
The molecular basis of AGE/RAGE activation in inducing skin autofluorescence, endothelial cell damage, and IL-6 gene expression in patients with SLE. HMGB-1: High mobility group box 1 protein; AGE: Advanced glycation end-product; RAGE: Receptor for AGE; MAPK: Mitogen activated protein kinase; *JUN*: Gene for ju-nana oncogene; NF-κB: Nuclear factor of kappa-light-chain-enhancer of activated B cells; EC: Endothelial cell.

**Figure 5 ijms-20-03878-f005:**
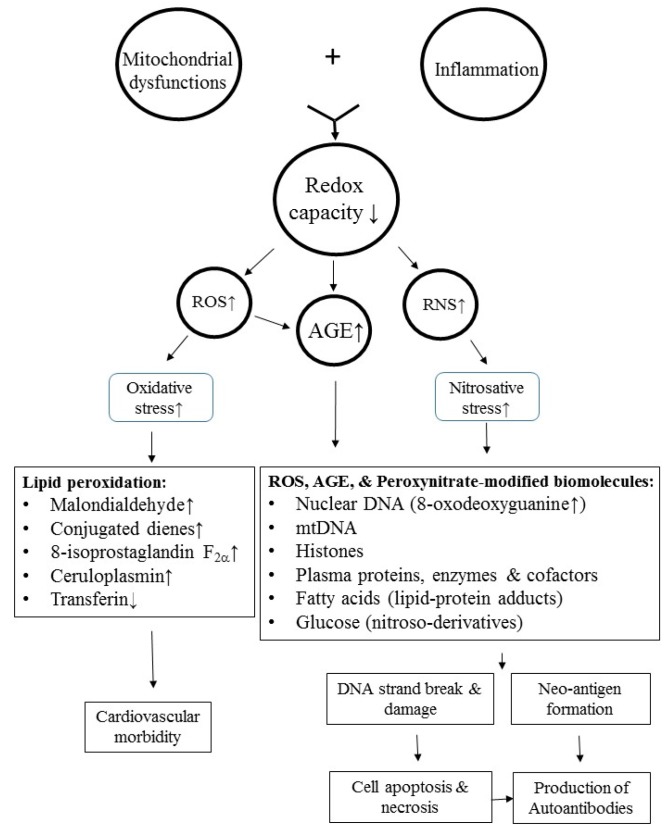
The molecular bases of mitochondrial dysfunction-elicited oxidative and nitrosative stresses in inducing autoantibody production and cardiovascular complications in patients with SLE. Redox: Reduction and oxidation; ROS: Reactive oxygen species; RNS: Reactive nitrogen species; AGE: Advanced glycation end-product; mt: Mitochondrial.

**Figure 6 ijms-20-03878-f006:**
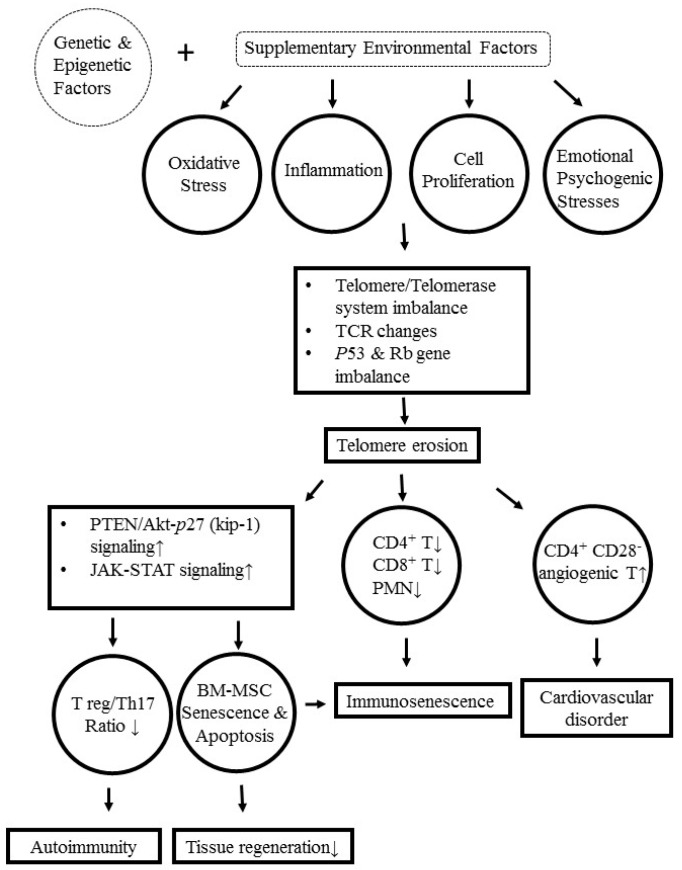
The molecular and cellular mechanisms underlying telomere/telomerase disequilibrium in inducing immunosenescence, increased CD4^+^CD28^−^ angiogenic T cell production, autoimmunity, and impaired tissue regeneration in patients with SLE. Broken lines are not discussed in the present review. BM-MSC: Bone marrow-derived mesenchymal stem cells; TCR: T cell receptor; PMN: Polymorphonuclear neutrophil: JAK-STAT: Janus kinase and signal transducer and activator of transcription; MW: Molecular weight; Treg: Regulatory T cell; PTEN: Phosphatase and tensin homolog; Akt: RAC-alpha serine/threonine-protein kinase; kip-1: Cyclin-dependent kinase inhibitor 1.

**Table 1 ijms-20-03878-t001:** Similarity in immune dysfunctions and common clinical features in physiological senescence and SLE.

Parameters		Physiological Senescence	SLE
**• Immunological Functions**			
Neutrophil:			
	Phagocytosis	↓ [8,14,15]	↓ [3]
	Chemotactic capacity	↓	↓
	Response to bacterial products (fMLP, LPS)	↓	↓
	Response to IL-8	↓	↓
	NETosis	ND	↑
	Link to Th1/Th2 cytokine ratio	ND	↓
Macrophage/dendritic cell:			↑
	Phagocytosis	↓ [8,16,17,18]	↓ [2,21,22]
	Chemotactic capacity	↑	↑
	Ability to stimulate lymphocyte	↑	↑
	Pro-inflammatory cytokine production	↑	↑
Natural killer cell:			
	Cytotoxicity	↓ [19]	↓ [20,23,24]
	Proliferation	↓	↓
T lymphocyte:			
	Th1	↓ [8,19]	↓ [25,26]
	Th2	↑	↑
	Th17	↑	↑
	Treg	↓	↓
	CD4^+^CD 28^null^ angiogenic T	↑	↑
	CD45RO^+^ T (memory T)	↑	↑
B lymphocyte:			
	CD5+ B	↑ [8,19]	↑ [26,27]
	Hypergammaglobulinemia	↑	↑
	ANA	↑	↑
	RFs	↑	↑
	APL	↑	↑
	ATA	↑	↑
**• Common clinical features:**			
	Infection rate	↑ [9,10,11,12,13]	↑ [2,3]
	Tumor incidence	↑	↑
	Cardiovascular diseases	↑	↑

fMLP: formyl-methionine-leucyl-phenylalanine; LPS: lipopolysaccharide; Treg: regulatory T cell; Th: helper T cell; ANA: antinuclear antibodies, RF: rheumatoid factor, APL: antiphospholipid antibodies; ATA: anti-thyroglobulin antibodies; CD45RO: marker for memory T cell; ND: no data.

**Table 2 ijms-20-03878-t002:** The cellular bases of inflammaging and its pathophysiological effects on patients with SLE.

Cellular Basis of Inflammaging	Pathophysiological Effects
• Decrease in the expression and function of TCR and its co-receptors for antigens in T cells [59]	Susceptible to infections
• Decrease in circulating B cells due to reduction of new B cell migration from bone marrow and B lymphopoiesis [60]	Antibody production ↓
•Shift from naïve to memory B cell [60] (naïve/memory B cell ratio ↓)	High affinity protective antibody production ↓
• Impaired ability of memory B cell differentiation to plasma cells [60]	Antibody production ↓
• CD4(+)CD28(+) angiogenic T cell ↓ [61,62,63] CD4(+)CD28(−) angiogenic T cell ↑	Endothelial cell damage ↑ Cardiovascular morbidity ↑
• Impaired IL-6/TGF-β balance [64,65]	Autoimmunity ↑, IL-22 ↑
• Th17 cell ↑	Inflammation ↑

TCR: T cell receptor; TGF: transforming growth factor; Th: helper T cell.

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
