# Peer review of "Molecular and Cellular Bases of Immunosenescence, Inflammation, and Cardiovascular Complications Mimicking “Inflammaging” in Patients with Systemic Lupus Erythematosus"

_ijms, 2019, doi:10.3390/ijms20163878_

Round 1
Reviewer 1 Report
The presented manuscript entitled „Molecular and Cellular Bases of Immunosenescence, Inflammation, and Cardiovascular Complications Mimicking “Inflamm-Aging” in Patients with Systemic Lupus Erythematosus” rises an interesting topic of immunosenescence aspects in SLE. Having been in this field for a few years, I find this review very interesting and worth being presented to the public.
In terms of the information presented in the manuscript, I see no major problems. Unfortunately, the style of the paper is uneven through the whole text and can cause some misunderstanding. Some parts of the text are written in a really good English and are easy to comprehend. On the other hand, other parts seem more incoherent and confusing (i.e. chapter 1, 2 or 6).
Therefore I advise a thorough language correction.
There are also a few other things making the manuscript difficult to read – mostly strange synonyms, sentences without a definite meaning. Let me list a few of them:
- Inflamm-aging – I know it is a bit picky, but I’d rather used “inflammaging” without a hyphen. Moreover, the Authors are not consequent in terms of using hyphens, as the same words are written either with or without hyphen throughout the text ( i.e. proinflammatory).
- The sequence of word in the sentence, i.e. “the factors involved in lupus pathogenesis are multiple including…” (abstract, page 1 line 19-20) – in my opinion should be: “there are multiple factors involved in lupus pathogenesis, including…” - but I am not a native speaker, thus I advise to use professional help.
- Page 7 lines 211-212 – “AGEs are a class of glycosylated macromolecules by non-enzymatic addition of saccharide derivatives.” This sentence should be rewritten in my opinion.
- Page 7 lines 220-221 – “ As the receptor for AGEs, RAGE belongs to the immunoglobulin superfamily of trans-membrane cell surface molecules” – I assume that the beginning of this sentence is unnecessary.
- Page 9 lines 272-273 – “Accordingly, peroxynitrite is a mixture of oxidant and
- 273 nitrative molecule in response to oxidative stress.” – is the verb missing?
There are more examples of confusing sentences throughout the text, that should be corrected.
- Synonyms – not being a native English speaker, I know how appealing is to find new, uncommon synonyms for ordinary words, but it can cause some dissonance. That is why I do advice to avoid really unusual words, i.e “stochastic”, “deranged”, “florid”, etc.
- Please be more consistent in explaining abbreviations. Regardless the list of abbreviations at the end of the manuscript, the Authors are inconsistent in explaining abbreviations throughout the text, i.e. GC-MS is not explained, while NMR, a few sentences later is explained and then put into brackets.
I would also like the Authors to go through the table 1 contents, as it seems that some citations are lacking. And please, make the figures a bit bigger, as they are illegible in the present form.
Summing up, I do find the presented manuscript suitable for publication in International Journal of Molecular Sciences. But the present form requires comprehensive language editing, therefore I suggest minor revision.
Author Response
Answers to the Reviewer (1):
Question (1): The style of the paper is uneven through the whole text and can cause misunderstanding. Some parts of the text are written in really good English and easy to comprehend. On the other hand, other parts seem more incoherent and confusing, i.e. chapter 1, 2 or 6.
Answer: Thanks for the important suggestion. We have revised these chapters extensively with rearrangement of their sequence and changes of the subtitles. We also sent the revised version for English editing as you have suggested to further improve the quality of writing. As you see, we have divided Chapter 2 into 2 parts, (2) and (3) so that the latter paragraph of the original (2) now becomes (3) to focus the factors contributing to “inflammaging”. In addition, we have made some revisions in figures 1 through 5 to correct the mistakes. Please see the respective revised paragraphs.
Question (2): There are also other things making the manuscript difficult to read- mostly strange synonyms, sentences without a definite meaning, for examples:
-“inflamm-aging” must be changed to “inflammaging”
-The authors are not consequent in terms of using hyphens, as the same words are written either with or without hyphen throughout the text (i.e. proinflammatory).
- The sequence of word in the sentence, i.e. in “Abstract, page 1 line 19-20; The factors involved in lupus pathogenesis are multiple including…” should be changed to “ There are multiple factors involved in lupus pathogenesis, including…”.
- Page 7 lines 211-212: AGEs are a class of glycosylated macromolecules by non-enzymatic addition of saccharide derivatives. This sentence should be rewritten.
-Page 7 lines 220-221: As the receptors for AGEs, RAGE belongs to the immunoglobulin superfamily of trans-membrane cell surface molecules. The beginning of the sentence is unnecessary.
- Page 9 lines 272-273 – “Accordingly, peroxynitrite is a mixture of oxidant and Lines 273 nitrative molecule in response to oxidative stress.” – is the verb missing?
-There are more examples of confusing sentences throughout the text that should be corrected.
- Synonyms: uncommon synonyms for ordinary words i.e “stochastic”, “deranged”, “florid”, etc.
Answer: Thanks for these valuable critiques. We have already corrected these inappropriate or strange “words” and “statements” in text for easy reading in the revised version. Please see the respective pages.
Question (3): Please be more consistent in explaining abbreviations. Regardless the list of abbreviations at the end of the manuscript, the Authors are inconsistent in explaining abbreviations throughout the text, i.e. GC-MS is not explained, while NMR, a few sentences later is explained and then put into brackets.
Answer: We have made explanations for these abbreviations, consistent to those appear for the 1st time in the text. The whole list of abbreviations are put in the last part of the revised version.
Question (4): I would also like the Authors to go through the table 1 contents, as it seems that some citations are lacking. And please, make the figures a bit bigger, as they are illegible in the present form.
Answer: Thanks for these reasonable requests. We have added several pertinent citations (Ref. 18, 22, 23, 26 and 27) in Table 1 according to the reviewer’s suggestions. We also enlarged the size of the words as bigger as we can in all figures to the extent that the original meaning of these cartoons are not distorted. New forms of all figures are submitted.
Question (5): The present form requires comprehensive language editing, therefore I suggest minor revision.
Answer: Thanks for the suggestion. We have followed your suggestion and have finished the English editing in the new version.

Reviewer 2 Report
This paper is timely, as it focuses on similar metabolic changes, such as GSH depletion and mTOR activation, in the immune system in aging and SLE. Important findings are omitted, such as that GSH depletion is linked to mitochondrial dysfunction and oxidative stress (Arthritis Rheum. 46:175-190) which can be reversed with therapeutic efficacy, using N-acetylcysteine or NAC (Arthritis Rheum. 64: 2937-2946). Interestingly, NAC also blocks mTOR activation which is a hallmark of aging within and outside the immune system (Ann. NY Acad. Sci. 1346:33-44). As recently unveiled, direct blockade of mTOR with rapamycin has remarkable clinical efficacy in SLE (Lancet, 391:1186-1196). This therapeutic effect of rapamycin involves the expansion of Tregs and effector-memory T cells which need to be included in Figure 6. mTOR activation has profound influence on lineage specification within the adaptive and innate immune systems (Trends Immunol. 39:562-576). NAC-responsive accumulation of kynurenine has been identified as a trigger of mTOR activation in SLE (Metabolomics. 11:1157-1174). Similar findings have been associated with aging (Front Aging Neurosci. 2018 Sep 6;10:263. doi: 10.3389/fnagi.2018.00263).
The above findings and considerations should be carefully incorporated into the review and mechanistic diagrams, which should markedly enhance the impact of the paper.
Author Response
Answers to Reviewer (2):
Question (1): This paper is timely, as it focuses on similar metabolic changes, such as GSH depletion and mTOR activation, in the immune system in aging and SLE. Important findings are omitted, such as that GSH depletion is linked to mitochondrial dysfunction and oxidative stress (Arthritis Rheum. 46:175-190) which can be reversed with therapeutic efficacy, using N-acetylcysteine or NAC (Arthritis Rheum. 64: 2937-2946). Interestingly, NAC also blocks mTOR activation which is a hallmark of aging within and outside the immune system (Ann. NY Acad. Sci. 1346:33-44). As recently unveiled, direct blockade of mTOR with rapamycin has remarkable clinical efficacy in SLE (Lancet, 391:1186-1196). This therapeutic effect of rapamycin involves the expansion of Treg and effector-memory T cells which need to be included in Figure 6. mTOR activation has profound influence on lineage specification within the adaptive and innate immune systems (Trends Immunol. 39:562-576). NAC-responsive accumulation of kynurenine has been identified as a trigger of mTOR activation in SLE (Metabolomics. 11:1157-1174). Similar findings have been associated with aging (Front Aging Neurosci. 2018 Sep 6;10:263. doi: 10.3389/fnagi.2018.00263).
The above findings and considerations should be carefully incorporated into the review and mechanistic diagrams, which should markedly enhance the impact of the paper.
Answer: Thank for this particularly important suggestion and the attached crucial references. We have added a new citation in page 5 (reference 72) and a new paragraph in page 8 (references 93-98) and cited these important references (references: 72, 93-98) to emphasize the interrelationships among mitochondria hyperpolarization, Oxidative stress, ATP depletion, mTOR activation and lupus pathogenesis. The paragraph is shown below:
“The cause of decreased GSH levels in SLE immune cells may result from mitochondrial hyperpolarization and increased reactive free radicals as reported by Huang and Perl [94]. Lai et al. [95] reported that N-acetylcystein, a GSH precursor, could reduce lupus disease activity by blocking mTOR in T cells. Perl et al. [96] further demonstrated in comprehensive metabolomics analyses that the accumulation of kynurenine in response to N-acetylcysteine in patients with SLE could become a biomarker for mTOR activation. In addition, rapamycin, the original mTOR blocker from which this adaptor protein was named, has been proved effective in the treatment of intractable SLE resistant to, or intolerant of conventional immunosuppressants as reported by Lai et al. [97]. Furthermore, Duval et al. [98] demonstrated that rapamycin treatment ameliorated age-related accumulation of toxic metabolites in brains of the mouse model of Down syndrome and aging.
In short conclusion, mTOR activation can determine the metabolic pathway in CD4+T cells of SLE patients to induce metabolic syndrome. These results are summarized in Figure 3.”

Round 2
Reviewer 2 Report
Extensive editing for English style and grammar is necessary.